# Substance use disorders in refugee and migrant groups in Sweden: A nationwide cohort study of 1.2 million people

Samantha Harris[1,2‡], Jennifer Dykxhoorn[1‡], Anna-Clara Hollander[3], Christina Dalman[3], James B. Kirkbride[1]*

**1** Psylife Group, Division of Psychiatry, University College London, London, United Kingdom, **2** Department of Psychosocial Science, University of Bergen, Bergen, Norway, **3** Department of Public Health Sciences, Karolinska Institute, Stockholm, Sweden

‡ These authors share first authorship on this work.
* j.kirkbride@ucl.ac.uk

**Data Availability Statement:** Data cannot be shared publicly under the terms and conditions of ethical approval for Psychiatry Sweden. This research had ethical approval as part of Psychiatry

## Abstract

### Background

Refugees are at higher risk of some psychiatric disorders, including post-traumatic stress disorder (PTSD) and psychosis, compared with other non-refugee migrants and the majority population. However, it is unclear whether this also applies to substance use disorders, which we investigated in a national register cohort study in Sweden. We also investigated whether risk varied by region of origin, age at migration, time in Sweden, and diagnosis of PTSD.

### Methods and findings

Using linked Swedish register data, we followed a cohort born between 1984 and 1997 from their 14th birthday or arrival in Sweden, if later, until an International Classification of Diseases, 10th revision (ICD-10), diagnosis of substance use disorder (codes F10.X–19.X), emigration, death, or end of follow-up (31 December 2016). Refugee and non-refugee migrants were restricted to those from regions with at least 1,000 refugees in the Swedish registers. We used Cox proportional hazards regression to estimate unadjusted and adjusted hazard ratios (aHRs) and 95% confidence intervals (CIs) in refugee and non-refugee migrants, compared with Swedish-born individuals, for all substance use disorders (F10.X–19.X), alcohol use disorders (F10.X), cannabis use disorders (F12.X), and polydrug use disorders (F19.X). In adjusted analyses, we controlled for age, sex, birth year, family income, family employment status, population density, and PTSD diagnosis. Our sample of 1,241,901 participants included 17,783 (1.4%) refugee and 104,250 (8.4%) non-refugee migrants. Refugees' regions of origin were represented in proportions ranging from 6.0% (Eastern Europe and Russia) to 41.4% (Middle East and North Africa); proportions of non-refugee migrants' regions of origin ranged from 11.8% (sub-Saharan Africa) to 33.7% (Middle East and North Africa). These groups were more economically disadvantaged at cohort entry (p < 0.001) than the Swedish-born population. Refugee (aHR: 0.52; 95% CI 0.46–0.60) and non-refugee (aHR: 0.46; 95% CI 0.43–0.49) migrants had similarly lower rates of

Sweden "Psykisk ohälsa, psykiatrisk sjukdom: förekomst och etiologi" (Mental health and psychiatric disorders–prevalence and aetiology), approved by the Stockholm Regional Ethical Review Board (number 2010/1185-31/5). Please contact the Stockholm Regional Ethical Review Board (https://etikprovningsmyndigheten.se/) for information about access to Swedish register data for researchers who meet the criteria for access to confidential data. Researchers can also contact the Swedish Research Council about how to access register-based data by visiting https://www.registerforskning.se/en/ or contacting them by email (registerforskning@vr.se) or telephone +46 8 546 440 00.

**Funding:** This work was supported by a Sir Henry Dale Fellowship jointly funded by the Wellcome Trust (www.wellcome.ac.uk) and the Royal Society (https://royalsociety.org/) (grant number: 101272/Z/13/Z to JBK), by a John Grace QC Scholarship from Mental Health Research UK (http://www.mentalhealthresearchuk.org.uk/) to JBK/JD, UCLH NIHR Biomedical Research Centre (https://www.uclh.nhs.uk/Research/BRC/) to SH, JD, and JBK, and by a UCL Overseas Research Scholarship (https://www.ucl.ac.uk) to JD. A-CH was supported by Folkhälsomyndigheten (Public Health Agency of Sweden) (https://www.folkhalsomyndigheten.se/) and FORTE (https://forte.se/en/), grant number 2017-00632. The funders had no role in study design, data collection and analysis, decision to publish, or preparation of the manuscript.

**Competing interests:** The authors have declared that no competing interests exist.

**Abbreviations:** aHR, adjusted hazard ratio; AIC, Akaike's Information Criterion; CI, confidence interval; HR, hazard ratio; ICD-10, International Classification of Diseases, 10th revision; IQR, interquartile range; LISA, Longitudinal Integration Database for Health Insurance and Labour Market Statistics; LRT, likelihood ratio test; PTSD, post-traumatic stress disorder.

all substance use disorders compared with Swedish-born individuals (crude incidence: 290.2 cases per 100,000 person-years; 95% CI 287.3–293.1). Rates of substance use disorders in migrants converged to the Swedish-born rate over time, indicated by both earlier age at migration and longer time in Sweden. We observed similar patterns for alcohol and polydrug use disorders, separately, although differences in cannabis use were less marked; findings did not differ substantially by migrants' region of origin. Finally, while a PTSD diagnosis was over 5 times more common in refugees than the Swedish-born population, it was more strongly associated with increased rates of substance use disorders in the Swedish-born population (aHR: 7.36; 95% CI 6.79–7.96) than non-refugee migrants (HR: 4.88; 95% CI 3.71–6.41; likelihood ratio test [LRT]: $p$ = 0.01). The main limitations of our study were possible non-differential or differential under-ascertainment (by migrant status) of those only seen via primary care and that our findings may not generalize to undocumented migrants, who were not part of this study.

## Conclusions

Our findings suggest that lower rates of substance use disorders in migrants and refugees may reflect prevalent behaviors with respect to substance use in migrants' countries of origin, although this effect appeared to diminish over time in Sweden, with rates converging towards the substantial burden of substance use morbidity we observed in the Swedish-born population.

## Author summary

### Why was this study done?

- Migrants and refugees have higher incidence rates of some psychiatric disorders, including post-traumatic stress disorder (PTSD) and psychotic disorders. It is unclear whether these groups face higher rates of substance use disorders than nonmigrant populations, or whether these rates change with time resident in a new country or due to comorbid PTSD.

- Most studies of substance use disorders in migrants and refugees have been restricted to small, cross-sectional surveys of prevalence, not incidence.

### What did the researchers do and find?

- We established a nationwide cohort study of over 1.2 million people aged up to 32 years old in Sweden, including over 17,000 refugees, to investigate incidence rates of substance use disorders diagnosed in secondary clinical care settings.

- The incidence rate of any substance use disorder, including alcohol and polydrug use disorders, was between 48% and 54% lower in refugees and non-refugee migrants from similar regions of origin than the Swedish-born population, who had particularly high rates of alcohol use disorders (208.4 new cases per 100,000 people per year; 95% CI 206.0–210.7). Differences in cannabis use disorders were less marked.

- For all outcomes, rates in migrants converged to the Swedish-born rate over time, indicated by earlier age at migration or longer time lived in Sweden.

- PTSD diagnoses were more common amongst refugees and non-refugee migrants than the Swedish-born population but were more strongly associated with risk of substance use disorders in the Swedish-born population.

### What do these findings mean?

- Prevalent behaviors with respect to substance abuse in refugees and migrants from particular regions may limit the likelihood of substance abuse and subsequent disorder, although these protective effects appear to diminish over time.

## Introduction

Over 258 million people, or 3.4% of the global population, currently live as migrants outside of their birth country [1], including over 25 million refugees [2] forcibly displaced by persecution, war, or violence. Refugees experience high rates of certain mental health disorders, including post-traumatic stress disorder (PTSD) [3–6] and psychotic disorders including schizophrenia [7]. Less epidemiological research has focused on the risk of other mental health disorders on refugees, although they appear to have lower risk of suicide than nonmigrant populations [8]. For non-refugee migrants, risk of depression and anxiety may be lower [9–12] or equivocal to autochthonous populations [11,13], but like refugees, they are at elevated risk of psychotic disorders [7,14–16].

The epidemiological evidence for risk of substance use disorders in refugee and non-refugee migrant groups is still poorly understood, with most studies focused on alcohol use behaviors and disorders [17]. Relatively few studies have compared rates of substance use disorders in refugees to non-refugee or nonmigrant populations, and available evidence presents a heterogeneous picture [17]. For example, one study in Sweden found that hospitalizations for alcohol and drug use disorders were lower in immigrants from regions of the world where Sweden has traditionally received substantial numbers of refugees, compared with the Swedish-born population [18]. This study, however, did not have a direct measure of refugee status. In contrast, another study in Sweden found that young male refugees were more likely to be hospitalized for drug-related problems [19]. Harmful alcohol consumption in Bhutanese refugees in Nepal was also no higher than typically observed in Western populations [20], while a further study in the US [21] reported that refugees were 3–6 times less likely to meet criteria for substance use disorders than the nonmigrant population. With regards to non-refugee migrant groups, current research also suggests that alcohol-related disorders are less common compared with host populations in the US [22], Scandinavia [23–25], and Spain [26,27], although not from one study in France [28]. Risk of other substance abuse disorders may also be lower in non-refugee migrants [22], although these have received less research attention to date.

There is also some evidence that the risk of some other psychiatric outcomes in migrant populations, most notably suicide [8], converge to the background Swedish rate with longer residence in Sweden. Moreover, earlier age at migration has been associated with increased

risk of psychotic disorders in some [29–31] (but not all [15]) studies, implicating possible cultural adaptation as a mechanism influencing future mental health. For substance use disorders, there is some corollary evidence from US data that substance use disorders are closer to native-born American prevalence in second-generation migrants and first-generation migrants with earlier age at migration [22], suggesting similar mechanisms; to our knowledge, this hypothesis has not been tested in incidence cohorts.

Many studies have been based on cross-sectional surveys [22,24–27], prevalence data [20,22,24,26,27], small samples [24,26,27], imprecise definitions of refugee status [18], or substance use behaviors rather than disorder [20,24,25], making it difficult to draw firm conclusions about the risk of substance use disorders in refugee and non-refugee migrant groups. Few population-based, longitudinal studies on this topic have been conducted, with rare exceptions [19]. However, this study did not disaggregate findings by substance type, while the increased risk of hospitalization for drug use disorders in male refugees contrasts the wider literature. One unexplored possibility here is that risk of substance use disorders in refugee and non-refugee migrant groups may vary by a third factor, such as exposure to traumatic life events or the comorbid experience of PTSD [32]. It has been suggested that over half of people with PTSD meet criteria for alcohol abuse, and over one third meet criteria for other substance abuse [33]. Therefore, exact substance use disorder risk in refugee and non-refugee migrants may depend on the comorbid presence of PTSD or may have a greater impact on substance use disorder rates, given that they are more likely to experience trauma and PTSD [34]. If true, we would expect that refugees, but not non-refugee migrants, would have a higher risk of substance use disorders than Swedish-born individuals and that the impact of PTSD (i.e., proportion exposed to PTSD) would be more common amongst refugee and non-refugee migrants. However, we would have no reason to believe that the effect of PTSD on substance abuse disorder risk (i.e., the relative risk or hazard ratio [HR]) would differ between refugees, non-refugees, and Swedish-born groups (i.e., no effect modification of the association between PTSD and rates of substance use disorders by migrant status). We therefore investigated whether the incidence of substance use disorders in refugees and non-refugee migrants from regions with substantial refugee flows to Sweden differed from the native-born population in a large, nationwide Swedish cohort, using high-quality linked register data. Specifically, we tested whether rates of substance use disorders would be lower in refugee and non-refugee migrant populations in Sweden; vary by region of origin; converge to the Swedish-born rate with earlier age at migration, and longer time lived in Sweden amongst migrant populations; and be independently associated with PTSD (that PTSD would be more common in refugee and non-refugee migrants [i.e., greater impact], but that PTSD would not modify the association between migrant status and substance use disorders).

## Methods

A prespecified protocol for this study was used (S1 Text) and is registered at protocols.io (dx.doi.org/10.17504/protocols.io.53fg8jn). We deviated from the protocol by including an analysis of time lived in Sweden in relation to the hazard of substance use disorders as a post hoc analysis.

### Study design and population

Using longitudinal Swedish register data from Psychiatry Sweden, a database of linked registers to explore the causes and outcomes of mental health disorders, we established an initial cohort of 1,345,320 people born between 1984 and 1997, of refugees, non-refugee migrants, and Swedish-born participants. We restricted the cohort to this birth period to ensure we

could follow all participants diagnosed with an International Classification of Diseases, 10th revision (ICD-10) substance use disorder (F10–F19), which was introduced in Sweden in 1997. We only included refugee and non-refugee migrants who arrived in Sweden from 1 January 1998 (when refugee status was first recorded in the longitudinal database for integration studies [STATIV]) from regions of origin where data were available in the Swedish registers on at least 1,000 refugees, to permit valid comparisons between these two groups, consistent with our previous methodology [7]. Migrants from other regions (e.g., Western Europe, the Americas, or Oceania) were excluded, as were children of migrants (second-generation migrants), defined as those born in Sweden to at least one foreign-born parent. Those not officially granted residence in Sweden (i.e., asylum seekers and undocumented migrants) and those diagnosed with non-affective psychosis (ICD-10 F20–F29) or a substance use disorder (F10–F19) before their 14th birthday were excluded. Participants were considered at risk of developing a substance use disorder from their 14th birthday (earliest: 1 January 1998) or arrival in Sweden, if later, and were followed until they received a diagnosis of a substance use disorder (see below), emigration, death, or the end of the follow-up period (31 December 2016).

## Outcomes

Our primary outcome of interest was a first ICD-10 diagnosis of mental and behavioral disorders due to psychoactive substance use (F10.X–F19.X) as recorded in the National Patient Register, following in- or outpatient admission. The National Patient Register began recording inpatient psychiatric contacts in 1973 and outpatient contacts in 2001 [35]. In- and outpatient coverages are known to be complete after 1987 and 2005, respectively [35], for publicly funded healthcare settings and privately funded inpatient care, and around 80% complete for privately funded outpatient settings. We also investigated three specific types of substance use disorders with considerable public health impact, as separate outcomes: alcohol use disorders (F10.X), cannabis use disorders (F12.X), and polydrug use disorders (F19.X). The earliest recorded date of diagnosis was used as the date of cohort exit.

## Exposures

Our primary exposure was migrant status, categorized as refugee, non-refugee migrant, or Swedish-born using information from the Total Population Register, STATIV, and the Multi-generational Register. Reason for settlement, including refugee status, was ascertained from the STATIV register, where individuals were coded as refugees if they had been granted a residency permit in accordance with Swedish law and the UN Refugee Convention as someone who, "owing to a well-founded fear of being persecuted [. . .] is unable to, or owing to such fear, is unwilling to avail himself of the protection of that country" [36]. All other immigrants granted official residency were classified as non-refugee migrants. Swedish-born people were defined as those born in Sweden to two Swedish-born parents, linked via the Multi-generational Register. As secondary exposures, we also considered region of origin, age at migration, and time in Sweden. We classified region of origin into 5 regions: Sweden; Eastern Europe and Russia; Asia; Middle East and North Africa; and sub-Saharan Africa. Full details of the specific region and countries of origin included in these definitions by the Swedish Migration Agency are reported in S1 Table. Age at migration was categorized as Swedish-born, 0–6 years, 7–15 years, 16–19 years, and 20+ years, consistent with age periods at which people in Sweden typically transition through the education system. Time in Sweden was categorized as Swedish-born, 0–4 years, 5–9 years, and 10+ years. Finally, we classified individuals as having a PTSD diagnosis if they had ever received an ICD-10 F43.1 diagnosis recorded in the National Patient Register.

## Confounders

Sex, birth year, family income, family employment status, population density, and PTSD diagnosis were included as confounders. We also treated age as a time-varying covariate, using Lexus expansion, to account for possible changes in substance use disorders risk for people as they aged. We created 5 age categories: 14–17, 18–21, 22–25, 26–29, and 30–32. We further adjusted for birth year to account for possible period effects. Family income was generated from the Longitudinal Integration Database for Health Insurance and Labour Market Statistics (LISA). As individuals are not included in the LISA until age 16, family income was used when available, which included highest parental income for parents residing at the same address as the participant. For individuals who entered the cohort after age 16, their personal income was utilized if parental income was not available. Family disposable income was divided into quintiles relative to the total Swedish population in a given year, which implicitly accounts for inflation over the follow-up period. Individuals were categorized according to the family income quintile in their year of cohort entry. Similarly, we considered parental employment status (employed versus unemployed), also retrieved from the LISA register. For migrants arriving after age 16 without parents, individual employment status was used. Population density was estimated for participants according to the "Small Area Marketing Statistic" neighbourhood (*N* = 9,209) in which they were registered in their year of cohort entry according to the Total Population Register. These neighbourhoods had a median population size of 726 in 2011 (interquartile range [IQR], 312–1,378). We generated four categories of population density: <25 people/km$^2$, 25.1–250 people/km$^2$, 250.1–2,500 people/km$^2$, and >2,500 people/km$^2$ to control for urban effects.

## Statistical analysis

We excluded participants with missing covariate data and compared them with the complete case sample. Next, we presented descriptive statistics on confounder and outcome variables, and reported incidence rates per 100,000 person-years by migrant status. We then fitted Cox proportional hazards models for each outcome, estimating unadjusted and adjusted hazard ratios (aHRs) and 95% confidence intervals (CIs) for each of our exposures. Due to multicollinearity between our exposures (all exposures had the Swedish-born population as the reference category), we fitted separate models for each exposure and reported Akaike's Information Criterion (AIC) for each, with lower score indicating better fit. We tested whether the effect of migrant status on substance use disorders was modified by PTSD, assessed formally via likelihood ratio tests (LRTs). To investigate possible biases introduced by excluding participants with missing covariate data, we reran our main unadjusted and adjusted Cox proportional hazards models for all substance use disorders, including people with missing income data (*N* = 99,631) in a sensitivity analysis; when controlling for income, we included those missing income as a separate category on this covariate. In a further sensitivity analysis restricted to migrant samples (i.e., excluding the Swedish-born reference category to reduce multicollinearity), we investigated whether migrant status, region of origin, age at migration, and time in Sweden had independent effects on risk of any substance use disorders. We tested the proportional hazards assumption for migrant status by conducting a proportional hazards test of departure of Schoenfeld residuals from zero over time, with inspection of log-log plots and by inspecting HRs by migrant status across the follow-up period in the event of violation of this assumption. We used Stata version 15 for all statistical analyses [37].

This study received ethical approval through Psychiatry Sweden from the Stockholm Regional Ethical Review Board (number 2010/1185-31/5) and consent was waived. We followed STROBE guidelines for reporting observational studies (S1 STROBE Checklist).

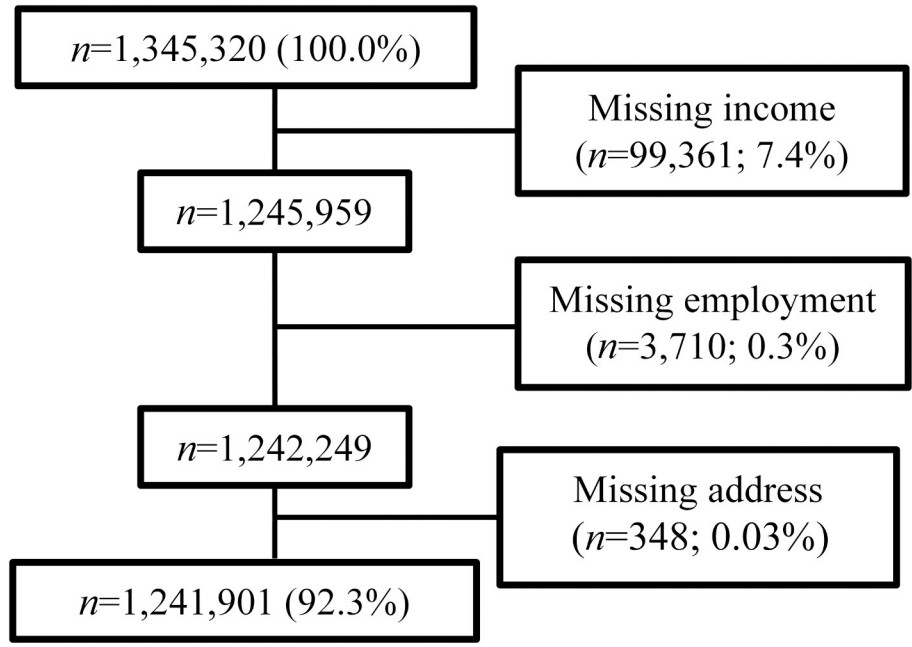

**Fig 1. PRISMA flow diagram.**

## Results

### Missing data

From 1,345,320 participants, we excluded 103,419 people (7.7%) missing covariate data (Fig 1), including family income ($n$ = 99,631; 7.4%), employment status ($n$ = 3,710; 0.3%), or address information in year of cohort entry ($n$ = 348; 0.03%). Participants with missing data were more likely to be refugee or non-refugee migrants, younger, male, from lower family income quintiles, unemployed families, and live in more urban areas than participants in the final analytical cohort (all $p < 0.001$; S2 Table). They were less likely to have received a diagnosis of any substance use disorder or PTSD (all $p < 0.001$; S2 Table).

### Demographic and clinical characteristics

Our final analytical sample included 1,241,901 people living in Sweden (92.3% of the total sample), including 17,783 refugees (1.4%), 104,250 non-refugee migrants (8.4%), and 1,119,868 Swedish-born individuals (90.2%) (**Table 1**). The largest proportion of refugee (41.4%) and non-refugee migrants (33.7%) were from the Middle East and North Africa (33.7%). The smallest proportion of refugee (6.0%) and non-refugee migrants (11.8%) were from Eastern Europe and Russia, and sub-Saharan Africa. Migrants from these broad regions tended to come from specific regions of origin from which Sweden has traditionally received substantial migrant flows, including East Africa, the former Yugoslavia, Central Asia (including Afghanistan), and Iraq (S1 Table). Refugee and non-refugee migrants were more likely to be from a lower family income category than the Swedish-born population, be older and be in a more densely populated part of Sweden at cohort entry (all $\chi^2$ $p < 0.001$). As expected, refugees were more likely to have a PTSD diagnosis (1.3%) than non-refugee migrants (0.6%) and the Swedish-born population (0.2%) ($\chi^2$ $p < 0.001$).

We identified 40,417 (2.9%) individuals who were first diagnosed with a substance use disorder in the inpatient or outpatient register over 14,439,302 person-years of follow-up

**Table 1. Cohort characteristics by migrant status in the complete case sample.**

| Characteristics | Swedish-born (*n* = 1,119,868; 90.2%) | | | Non-refugee migrants (*n* = 104,250; 8.4%) | | | Refugees (*n* = 17,783; 1.4%) | | |
|---|---|---|---|---|---|---|---|---|---|
| | *N* | Percent | Person-years | *N* | Percent | Person-years | *N* | Percent | Person-years |
| **Psychiatric diagnosis** | | | | | | | | | |
| Any substance use disorder | 38,971 | 3.5 | 13,427,338 | 1,223 | 1.2 | 862,574 | 223 | 1.3 | 149,389 |
| Alcohol use disorder | 29,444 | 2.6 | 13,498,936 | 690 | 0.7 | 865,865 | 117 | 0.7 | 150,000 |
| Cannabis use disorder | 4,381 | 0.4 | 13,730,555 | 301 | 0.3 | 868,961 | 52 | 0.3 | 150,534 |
| Polydrug use disorder | 7,553 | 0.7 | 13,704,514 | 246 | 0.2 | 869,319 | 53 | 0.3 | 150,501 |
| **Sex** | | | | | | | | | |
| Female | 545,163 | 48.7 | 6,514,361 | 53,104 | 50.9 | 442,936 | 7,798 | 43.8 | 65,542 |
| Male | 574,705 | 51.3 | 6,912,978 | 51,146 | 49.1 | 419,638 | 9,985 | 56.2 | 83,848 |
| **Birth year** | | | | | | | | | |
| 1984–1985 | 155,362 | 13.9 | 2,642,047 | 28,029 | 26.9 | 231,807 | 3,613 | 20.3 | 32,313 |
| 1986–1987 | 166,546 | 14.9 | 2,523,494 | 21,851 | 21.0 | 181,848 | 3,063 | 17.2 | 27,592 |
| 1998–1999 | 182,803 | 16.3 | 2,436,898 | 15,632 | 15.0 | 138,784 | 2,784 | 15.7 | 24,706 |
| 1990–1991 | 195,903 | 17.5 | 2,261,732 | 13,419 | 12.9 | 122,323 | 2,709 | 15.2 | 23,664 |
| 1992–1993 | 184,705 | 16.5 | 1,804,263 | 8,640 | 8.3 | 69,895 | 2,485 | 14.0 | 18,977 |
| 1993–1995 | 163,036 | 14.6 | 1,294,891 | 11,198 | 10.7 | 83,404 | 2,255 | 12.7 | 16,556 |
| 1996–1997 | 71,513 | 6.4 | 464,013 | 5,481 | 5.3 | 34,514 | 874 | 4.9 | 5,582 |
| **Region of origin** | | | | | | | | | |
| Sweden | 1,119,868 | 100.0 | 13,427,338 | | | | | | |
| Eastern Europe and Russia | | | | 30,236 | 29.0 | 246,832 | 1,066 | 6.0 | 10,770 |
| Asia | | | | 26,582 | 25.5 | 197,954 | 3,063 | 17.2 | 25,888 |
| Middle East and North Africa | | | | 35,088 | 33.7 | 313,406 | 7,363 | 41.4 | 65,936 |
| Sub-Saharan Africa | | | | 12,344 | 11.8 | 104,382 | 6,291 | 35.4 | 46,796 |
| **Age at migration** | | | | | | | | | |
| Sweden | 1,119,868 | 100.0 | 13,427,338 | | | | | | |
| 0–6 years | | | | 5,115 | 4.9 | 37,250 | 1,014 | 5.7 | 7,423 |
| 7–15 years | | | | 35,314 | 33.9 | 348,804 | 5,175 | 29.1 | 49,216 |
| 16–19 years | | | | 21,761 | 20.9 | 198,099 | 5,219 | 29.4 | 44,114 |
| 20+ years | | | | 42,060 | 40.4 | 278,422 | 6,375 | 35.9 | 48,635 |
| **Time in Sweden** | | | | | | | | | |
| Sweden | 1,119,868 | 100.0 | 13,427,338 | | | | | | |
| 0–4 years | | | | 94,733 | 90.9 | 787,642 | 16,196 | 91.1 | 136,937 |
| 5–9 years | | | | 8,218 | 7.9 | 66,244 | 1,214 | 6.8 | 10,037 |
| 10+ years | | | | 1,299 | 1.3 | 8,688 | 373 | 2.1 | 2,414 |
| **Family income** | | | | | | | | | |
| Quintile 1 (lowest) | 11,021 | 1.0 | 121,140 | 57,300 | 55.0 | 420,470 | 12,213 | 68.7 | 98,150 |
| Quintile 2 | 77,101 | 6.9 | 904,388 | 19,242 | 18.5 | 176,311 | 1,998 | 11.2 | 17,842 |
| Quintile 3 | 204,960 | 18.3 | 2,452,882 | 18,047 | 17.3 | 173,216 | 2,429 | 13.7 | 22,416 |
| Quintile 4 | 397,382 | 35.5 | 4,782,866 | 7,146 | 6.8 | 69,198 | 893 | 5.0 | 8,590 |
| Quintile 5 (highest) | 429,404 | 38.3 | 5,166,062 | 2,515 | 2.4 | 23,379 | 250 | 1.4 | 2,391 |
| **Family employment** | | | | | | | | | |
| Unemployed | 197,330 | 17.6 | 2,443,022 | 4,372 | 4.2 | 39,224 | 501 | 2.8 | 4,347 |
| Employed | 922,538 | 82.4 | 10,984,316 | 99,878 | 95.8 | 823,350 | 17,282 | 97.2 | 145,042 |
| **Population density** | | | | | | | | | |
| 0–25 pp/km$^2$ | 280,679 | 25.1 | 3,415,094 | 10,553 | 10.1 | 93,166 | 5,202 | 29.3 | 43,615 |
| 25.1–250 pp/km$^2$ | 616,654 | 55.1 | 7,398,665 | 48,239 | 46.3 | 401,105 | 8,584 | 48.3 | 72,387 |

*(Continued)*

**Table 1.** (*Continued*)

| Characteristics | Swedish-born (*n* = 1,119,868; 90.2%) | | | Non-refugee migrants (*n* = 104,250; 8.4%) | | | Refugees (*n* = 17,783; 1.4%) | | |
|---|---|---|---|---|---|---|---|---|---|
| | *N* | Percent | Person-years | *N* | Percent | Person-years | *N* | Percent | Person-years |
| 250.1–2,500 pp/km$^2$ | 161,798 | 14.5 | 1,903,867 | 30,385 | 29.2 | 244,110 | 2,663 | 15.0 | 22,204 |
| 2,500.1 or more pp/km$^2$ | 60,737 | 5.4 | 709,712 | 15,073 | 14.5 | 124,196 | 1,334 | 7.5 | 11,184 |
| **PTSD diagnosis** | 2,370 | 0.2 | 25,886 | 659 | 0.6 | 6,607 | 227 | 1.3 | 1,866 |

Abbreviations: pp/km$^2$, people per square kilometer; PTSD, post-traumatic stress disorder

(**Table 1**). Both refugee (19.8 years; IQR, 17.4–22.8) and non-refugee migrants (19.9 years; IQR, 17.5–22.7) had an older median age of first diagnosis than Swedish-born participants (18.6 years; IQR, 16.5–20.9; Kruskal-Wallis: $\chi^2(2)$ = 189.2; $p$ = 0.0001).

## Rates of any substance use disorder by migrant status

The crude incidence of substance use disorders was higher in the Swedish-born population (290.2 per 100,000 person-years; 95% CI 287.3–293.1) than non-refugee migrants (141.8 per 100,000 person-years; 95% CI 134.0–150.0) and refugees (149.3 per 100,000 person-years, 95% CI 130.9–170.2). In an unadjusted model, both refugee (HR, 0.42; 95% CI 0.37–0.48; $p < 0.001$) and non-refugee (HR, 0.42; 95% CI 0.39–0.44; $p < 0.001$) migrants were less likely to have received a diagnosis for a substance use disorder than the Swedish-born population (**Table 2**),

**Table 2. Unadjusted and adjusted HRs by migrant status.**

| Substance use disorder | *N* | Percent | Unadjusted | | | | Adjusted | | | *p*-value |
|---|---|---|---|---|---|---|---|---|---|---|
| | | | HR | 95% CI | | *p*-value | HR[1] | 95% CI | | |
| **Any substance use disorder** | | | | | | | | | | |
| Swedish-born | 38,971 | 3.5 | 1 | | | | 1 | | | |
| Non-refugee migrants | 1,223 | 1.2 | 0.42 | 0.39 | 0.44 | <0.001 | 0.46 | 0.43 | 0.49 | <0.001 |
| Refugees | 223 | 1.3 | 0.42 | 0.37 | 0.48 | <0.001 | 0.52 | 0.46 | 0.60 | <0.001 |
| Refugees versus non-refugee migrants | - | - | 1.02 | 0.88 | 1.18 | 0.78 | 1.15 | 0.99 | 1.32 | 0.06 |
| **Alcohol use disorder** | | | | | | | | | | |
| Swedish-born | 29,444 | 2.6 | 1 | | | | 1 | | | |
| Non-refugee migrants | 690 | 0.7 | 0.30 | 0.28 | 0.33 | <0.001 | 0.35 | 0.32 | 0.39 | <0.001 |
| Refugees | 117 | 0.7 | 0.29 | 0.24 | 0.34 | <0.001 | 0.38 | 0.31 | 0.46 | <0.001 |
| Refugees versus non-refugee migrants | - | - | 0.95 | 0.78 | 1.15 | 0.58 | 1.06 | 0.87 | 1.30 | 0.54 |
| **Cannabis use disorder** | | | | | | | | | | |
| Swedish-born | 4,381 | 0.4 | 1 | | | | 1 | | | |
| Non-refugee migrants | 301 | 0.3 | 0.96 | 0.85 | 1.08 | 0.49 | 0.81 | 0.70 | 0.93 | 0.003 |
| Refugees | 52 | 0.3 | 0.93 | 0.71 | 1.22 | 0.59 | 0.96 | 0.72 | 1.29 | 0.78 |
| Refugees versus non-refugee migrants | - | - | 0.97 | 0.72 | 1.30 | 0.92 | 1.18 | 0.88 | 1.59 | 0.27 |
| **Polydrug use disorder** | | | | | | | | | | |
| Swedish-born | 7,553 | 0.7 | 1 | | | | 1 | | | |
| Non-refugee migrants | 246 | 0.2 | 0.48 | 0.43 | 0.55 | <0.001 | 0.44 | 0.37 | 0.51 | <0.001 |
| Refugees | 53 | 0.3 | 0.59 | 0.45 | 0.77 | <0.001 | 0.58 | 0.44 | 0.78 | <0.001 |
| Refugees versus non-refugee migrants | - | - | 1.21 | 0.90 | 1.63 | 0.21 | 1.34 | 0.99 | 1.80 | 0.06 |

[1]Adjusted for age, sex, birth year, family income, family employment, population density, and PTSD diagnosis.

Abbreviations: CI, confidence interval; HR, hazard ratio

patterns which persisted following adjustment for age, sex, birth year, family income, family employment, population density, and PTSD diagnosis (HR$_{refugees}$, 0.52; 95% CI 0.46–0.60; $p < 0.001$; HR$_{migrants}$, 0.46; 95% CI 0.43–0.49; $p < 0.001$).

## Rates of specific substance use disorders by migrant status

Alcohol use disorders were the most commonly diagnosed substance use disorder in our cohort (208.4 per 100,000 person-years; 95% CI 206.0–210.7). As for all substance use disorders, rates of alcohol use disorder were lower in both refugee (HR: 0.38; 95% CI 0.31–0.46; $p < 0.001$) and non-refugee migrants (HR: 0.35; 95% CI 0.32–0.39; $p < 0.001$) after adjustment for confounders (**Table 2**). Polydrug use disorders showed similar patterns by migrant status. Interestingly, cannabis use disorders initially appeared not to be reduced in refugee and non-refugee migrants relative to the Swedish-born population in an unadjusted model (i.e., HR$_{refugees}$: 0.93; 95% CI 0.71–1.22; $p = 0.59$). After adjustment for confounders, there remained no evidence that refugees were at lower risk (HR: 0.96; 95% CI 0.72–1.19; $p = 0.78$), although non-refugee migrants were at slightly lower risk of cannabis use disorders than the Swedish-born population (HR: 0.81; 95% CI 0.70–0.93; $p = 0.003$). There were no differences in rates of alcohol or cannabis use disorders between refugee and non-refugee migrants (i.e., alcohol use disorders, HR: 1.06; 95% CI 0.87–1.30; $p = 0.54$), although there was weak evidence that poly-drug use disorders may have been somewhat higher in refugees than non-refugee migrants (HR: 1.34; 95% CI 0.99–1.80; $p = 0.06$). In sensitivity analyses in a larger cohort of 1.34 million participants, including 99,631 missing data on income, there was little evidence that excluding this group biased our findings (S5 Table).

When compared with Swedish-born participants, migrants (both refugee and non-refugee migrants) from all regions of origin were less likely to receive a diagnosis of any substance use disorder, with similar magnitudes of reduced risk for participants from all regions of origin (**Table 3**). Similar patterns were found for alcohol and polydrug use disorders. We found reduced risk of cannabis use disorders for people from most regions of origin, except for migrants from sub-Saharan Africa, who were more likely to be diagnosed with these disorders (HR: 1.40; 95% CI 1.11–1.76; $p = 0.004$) than the Swedish-born population.

## Rates of substance use disorders by age at migration and time in Sweden

For all outcomes, we found dose-response relationships between rates of substance use disorders and earlier age at migration or longer time in Sweden (**Table 4** and S4 Table), such that HRs tended to converge to the Swedish-born rate over time. For example, for any substance use disorder, the aHR was 0.24 (95% CI 0.18–0.31; $p < 0.001$) for those who immigrated to Sweden after 20 years old, but 0.73 (95% CI 0.60–0.89; $p < 0.001$) in those who immigrated between 0 and 6 years old (**Table 4**). For cannabis use disorders (HR: 0.94; 95% CI 0.62–1.42; $p = 0.77$) and polydrug use disorders (HR: 0.92; 95% CI 0.59–1.44; $p = 0.73$), we found no statistically significant differences in rates between the Swedish-born population and those who immigrated between 0 and 6 years old (S4 Table). These dose-response patterns were mirrored by time in Sweden, with no statistically significant differences in rates of any substance use disorder outcome between the Swedish-born population and migrant groups who had lived in Sweden for over 10 years (**Table 4**, S4 Table). Inspection of AIC scores suggested that age at migration provided better fit of the data than any other exposure for any substance use disorders and alcohol use disorders (jointly with time in Sweden; S5 Table), while region of origin provided a better fit for cannabis and polydrug use disorders.

In further analyses of any substance use disorder, excluding the Swedish-born group to reduce multicollinearity between our exposures, we were able to mutually adjust for migrant status,

region of origin, age at migration, and time in Sweden (S6 Table), which suggested independent effects for the latter three variables on rates of substance use disorder amongst migrant groups.

## Rates of substance use disorders by PTSD and migrant status

There was evidence (Table 5) that the association between PTSD and all substance use disorders differed by migrant status (i.e., LRT for interaction between PTSD and migrant status for any substance use disorder: $\chi^2$ on 2 degrees of freedom ($df$): 9.4; $p = 0.01$). Typically, PTSD was associated with a larger risk of substance use disorders in the Swedish-born population than in non-refugee migrants or refugee migrants. This effect was most pronounced for cannabis use disorders (LRT $\chi^2$ (2): 11.7; $p = 0.003$), such that PTSD appeared to be associated with greater risk of cannabis use disorders in the Swedish-born population (HR: 11.50; 95% CI 9.30–14.23; $p < 0.001$) than non-refugee (HR: 5.92; 95% CI 3.58–9.81; $p < 0.001$) or refugee migrants (HR: 2.60; 95% CI 0.63–10.69; $p = 0.19$).

## Assumptions

For any substance use disorder, there was some evidence of violation of the proportional hazards assumption for our migrant status variable. This effect was only apparent for non-refugee

**Table 3. Unadjusted and adjusted HRs by region of origin.**

| Substance use disorder | N | Percent | Unadjusted | | | | Adjusted | | | p-value |
|---|---|---|---|---|---|---|---|---|---|---|
| | | | HR | 95% CI | | p-value | HR[1] | 95% CI | | |
| **Any substance use disorder** | | | | | | | | | | |
| Swedish | 38,971 | 3.5 | 1 | | | | 1 | | | |
| Eastern Europe and Russia | 391 | 1.3 | 0.44 | 0.40 | 0.49 | <0.001 | 0.51 | 0.45 | 0.56 | <0.001 |
| Asia | 266 | 0.9 | 0.34 | 0.30 | 0.38 | <0.001 | 0.42 | 0.37 | 0.48 | <0.001 |
| Middle East and North Africa | 532 | 1.3 | 0.42 | 0.38 | 0.45 | <0.001 | 0.43 | 0.39 | 0.47 | <0.001 |
| Sub-Saharan Africa | 257 | 1.4 | 0.48 | 0.43 | 0.55 | <0.001 | 0.57 | 0.50 | 0.65 | <0.001 |
| **Alcohol use disorder** | | | | | | | | | | |
| Swedish | 29,444 | 2.6 | 1 | | | | 1 | | | |
| Eastern Europe and Russia | 222 | 0.7 | 0.33 | 0.28 | 0.37 | <0.001 | 0.40 | 0.35 | 0.46 | <0.001 |
| Asia | 175 | 0.6 | 0.28 | 0.24 | 0.33 | <0.001 | 0.38 | 0.32 | 0.45 | <0.001 |
| Middle East and North Africa | 269 | 0.6 | 0.27 | 0.24 | 0.31 | <0.001 | 0.30 | 0.27 | 0.34 | <0.001 |
| Sub-Saharan Africa | 141 | 0.8 | 0.34 | 0.29 | 0.40 | <0.001 | 0.42 | 0.35 | 0.50 | <0.001 |
| **Cannabis use disorder** | | | | | | | | | | |
| Swedish | 4,381 | 0.4 | 1 | | | | 1 | | | |
| Eastern Europe and Russia | 77 | 0.3 | 0.82 | 0.66 | 1.03 | 0.09 | 0.76 | 0.60 | 0.97 | 0.03 |
| Asia | 49 | 0.2 | 0.59 | 0.45 | 0.79 | <0.001 | 0.55 | 0.41 | 0.74 | <0.001 |
| Middle East and North Africa | 141 | 0.3 | 1.03 | 0.87 | 1.22 | 0.73 | 0.79 | 0.66 | 0.95 | 0.01 |
| Sub-Saharan Africa | 86 | 0.5 | 1.53 | 1.23 | 1.89 | <0.001 | 1.40 | 1.11 | 1.76 | 0.004 |
| **Poly-drug use disorder** | | | | | | | | | | |
| Swedish | 7,553 | 0.7 | 1 | | | | 1 | | | |
| Eastern Europe and Russia | 79 | 0.3 | 0.52 | 0.41 | 0.65 | <0.001 | 0.45 | 0.35 | 0.57 | <0.001 |
| Asia | 34 | 0.1 | 0.26 | 0.18 | 0.36 | <0.001 | 0.26 | 0.18 | 0.37 | <0.001 |
| Middle East and North Africa | 113 | 0.3 | 0.50 | 0.42 | 0.60 | <0.001 | 0.42 | 0.35 | 0.52 | <0.001 |
| Sub-Saharan Africa | 73 | 0.4 | 0.81 | 0.64 | 1.01 | 0.07 | 0.85 | 0.66 | 1.10 | 0.22 |

[1]Adjusted for age, sex, birth year, family income, family employment, population density, and PTSD diagnosis.

Abbreviations: CI, confidence interval; HR, hazard ratio

**Table 4. Unadjusted and adjusted HRs by age at migration and time in Sweden for any substance use disorder.**

| Substance use disorder | N | Percent | Unadjusted | | | Adjusted | | | |
| --- | --- | --- | --- | --- | --- | --- | --- | --- | --- |
| | | | HR | 95% CI | | *p*-value | HR[1] | 95% CI | | *p*-value |
| **Age at migration** | | | | | | | | | |
| Swedish-born | 38,971 | 3.5 | 1 | | | | 1 | | | |
| 0–6 years | 105 | 1.7 | 0.64 | 0.53 | 0.77 | <0.001 | 0.73 | 0.60 | 0.89 | 0.002 |
| 7–15 years | 823 | 2.0 | 0.64 | 0.60 | 0.69 | <0.001 | 0.48 | 0.45 | 0.52 | <0.001 |
| 16–19 years | 323 | 1.2 | 0.39 | 0.35 | 0.44 | <0.001 | 0.28 | 0.24 | 0.33 | <0.001 |
| 20+ years | 195 | 0.4 | 0.16 | 0.14 | 0.19 | <0.001 | 0.24 | 0.18 | 0.31 | <0.001 |
| **Time in Sweden** | | | | | | | | | |
| Swedish-born | 38,971 | 3.5 | 1 | | | | 1 | | | |
| 0–4 years | 1,234 | 1.1 | 0.40 | 0.37 | 0.41 | <0.001 | 0.42 | 0.39 | 0.46 | <0.001 |
| 5–9 years | 170 | 1.8 | 0.62 | 0.54 | 0.73 | <0.001 | 0.65 | 0.56 | 0.76 | <0.001 |
| 10+ years | 42 | 2.5 | 1.03 | 0.76 | 1.39 | 0.86 | 1.11 | 0.82 | 1.51 | 0.49 |

[1]Adjusted for age, sex, birth year, family income, family employment, population density, and PTSD diagnosis.

Abbreviations: CI, confidence interval; HR, hazard ratio

migrants (S1 Fig), primarily driven by slightly higher incidence rates of substance use disorders amongst non-refugee migrants in later follow-up periods (S7 Table). Nevertheless, HRs for both migrant groups were substantially lower than for the Swedish-born group across all follow-up periods (S7 Table).

**Table 5. Effect modification between migrant status and PTSD diagnosis in adjusted models.**

| Substance use disorder | N[1] | Percent | Adjusted | | | *p*-value |
| --- | --- | --- | --- | --- | --- | --- |
| | | | HR[2] | 95% CI | | |
| **Any substance use disorder** | | | | | | |
| PTSD in Swedish-born | 627 | 26.5 | 7.36 | 6.79 | 7.96 | <0.001 |
| PTSD in nonmigrant refugees | 54 | 8.2 | 4.88 | 3.71 | 6.41 | <0.001 |
| PTSD in refugees | 17 | 7.5 | 5.94 | 3.62 | 9.74 | <0.001 |
| *LRT χ² (df) p-value for interaction* | 9.4 (2); *p* = 0.01 | | | | | |
| **Alcohol use disorder** | | | | | | |
| PTSD in Swedish-born | 410 | 17.3 | 5.93 | 5.38 | 6.54 | <0.001 |
| PTSD in nonmigrant refugees | 23 | 2.5 | 3.60 | 2.37 | 5.45 | <0.001 |
| PTSD in refugees | 6 | 2.6 | 3.82 | 1.68 | 8.68 | <0.001 |
| *LRT χ² (df) p-value for interaction* | 7.1 (2); *p* = 0.03 | | | | | |
| **Cannabis use disorder** | | | | | | |
| PTSD in Swedish-born | 88 | 3.7 | 11.50 | 9.30 | 14.23 | <0.001 |
| PTSD in nonmigrant refugees | 16 | 2.4 | 5.92 | 3.58 | 9.81 | <0.001 |
| PTSD in refugees | 2 | 0.9 | 2.60 | 0.63 | 10.69 | 0.19 |
| *LRT χ² (df) p-value for interaction* | 11.7 (2); *p* = 0.003 | | | | | |
| **Polydrug use disorder** | | | | | | |
| PTSD in Swedish-born | 238 | 10.0 | 13.83 | 12.14 | 15.76 | <0.001 |
| PTSD in nonmigrant refugees | 18 | 2.7 | 7.10 | 4.39 | 11.48 | <0.001 |
| PTSD in refugees | 9 | 4.0 | 13.77 | 6.72 | 28.22 | <0.001 |
| *LRT χ² (df) p-value for interaction* | 8.1 (2); *p* = 0.02 | | | | | |

[1]Number and proportion of participants with PTSD diagnosis who received a substance use disorder diagnosis.

[2]Adjusted for age, sex, birth year, family income, family employment, and population density.

Abbreviations: CI, confidence interval; df, degrees of freedom; HR, hazard ratio; LRT, likelihood ratio test

## Discussion

### Primary findings

In our large, nationwide cohort, refugee and non-refugee migrants were substantially less likely to be diagnosed with a substance use disorder than Swedish-born individuals, extending to alcohol, cannabis (in non-refugee migrants), and polydrug disorders independently. We found no evidence of differences in incidence rates between these two migrant groups, with the possible exception of polydrug use disorders, which may have been elevated amongst refugees compared with non-refugee migrants. Rates were lower for migrants from all regions of origin, with the exception of raised rates of cannabis use disorders for migrants from sub-Saharan Africa relative to the Swedish-born population. Rates in migrants converged to the Swedish-born rate over time, with dose-response patterns for both age at migration and time in Sweden. Finally, we found strong evidence that individuals with PTSD were more likely to be diagnosed with a substance use disorder than those without a PTSD diagnosis, a relative effect that was more strongly associated with risk amongst Swedish-born participants, in contrast to our hypothesis. Nonetheless, PTSD may have had greater impact on the incidence of substance abuse disorders in migrant groups, given its higher occurrence than in the Swedish-born population.

### Strengths and limitations

Our study had several methodological strengths. Our longitudinal design in a comprehensive nationwide sample allowed us to obtain precise estimates of the incidence of substance use disorders in refugees and non-refugee migrants in comparison with the Swedish-born population for the first time, to our knowledge. Because of the unique opportunity offered by the linked Swedish registers, we were able to gather information on over 1.2 million people living in Sweden, including over 17,000 refugees. The Swedish register is known to be reliable for research, containing highly complete data with minimal loss to follow-up [7]. We conducted complete case analysis, excluding 7.7% of the cohort due to missing covariate data. Excluded participants predominantly left the cohort before age 16 (S2 Table), on whom personal or parental income was not available in the LISA register before this age. Although these participants differed from the analytical sample on outcome, exposure, and covariate data, sensitivity analyses that retained these participants did not alter the interpretation of our findings (S3 Table). Furthermore, the literature also suggests that missingness of around 5% is likely to be inconsequential [38] to valid statistical inferences and that estimates are unlikely to be biased with less than 10% missing data [39,40]. Rates of all-cause mortality in this young sample were low (S8 Table) and were therefore unlikely to have acted as a competing risk on our results. Together, this suggests that our findings were robust to possible biases introduced by their omission from our main analyses.

There were also notable limitations of our study. Most crucially, our incidence results were based on substance use disorders diagnosed in secondary care, including emergency department visits but excluding contacts only seen within primary care. Here, non-differential case ascertainment would have led us to underestimate the incidence of substance use disorders in Sweden, particularly where certain patterns of misuse and related disorders (i.e., alcohol) may go undetected in the population. Nonetheless, we would still interpret our results as informative of the treated incidence of such disorders requiring secondary in- or outpatient admission. In this light, the absolute incidence rates we observed highlight a substantive public mental health issue, particularly amongst the Swedish-born population, in whom the overall incidence rate of 290 new cases per 100,000 person-years is seven and a half times higher than reported for a near-identical cohort with respect to psychotic disorders [7]. Furthermore, these rates

appear consistent with those observed in similar comprehensive national data in Denmark [14] and substantively higher than reported in general population samples in the Netherlands [41] and US [42]. We acknowledge that differential case ascertainment by migrant status may have had more substantive impact on our results. This could have occurred if migrant groups were more or less likely to only be identified via primary care. It has previously been reported that refugees in Sweden struggle to access healthcare services [43], sometimes due to a fear of being deported [27]. Other obstacles to accessing healthcare may include language, legal, gender, or stigma-based barriers [44]. If true, this may have led us to underestimate rates of substance use disorders in refugee migrants relative to the Swedish-born population. Nevertheless, one might expect some of these issues to apply less readily to non-refugee migrants, whose lower risk of substance abuse disorders was equivocal to their refugee counterparts from the same regions of origin. Furthermore, such an ascertainment bias is difficult to reconcile with the substantively raised rates of PTSD and psychotic disorders observed in refugees from similar samples in Sweden [7,45]. Although a small proportion of cases presenting to private outpatient facilities may have been missed, we believe such an effect would have led our estimates to be conservative, assuming that refugee and non-refugee migrants in our sample would have been less likely to use private healthcare. Although the National Patient Register only included outpatient psychiatric diagnoses recorded since 2001 (with complete national coverage since 2005), we do not believe this would have led to substantial case ascertainment bias, given the paucity of outpatient services available before 2001 in Sweden.

Our results should also be interpreted in the context of the validity of diagnoses made in clinical practice. In general, psychiatric diagnoses in the National Patient Register are valid [35], although direct evidence for substance use disorders remains to be established. Nonetheless, diagnoses in our study were based on those made following inpatient admission or specialized outpatient treatment, and we have no reason to believe they would not be valid.

A further limitation of our study is that we used broad region of origin categories, although our migrant samples tended to come from specific regions from these areas (S1 Table). Our findings may not generalize to asylum seekers or undocumented migrants who were not included in the present study. There was some evidence that hazards were not proportional for refugee and non-refugee migrants, although these groups were at lower risk of substance use disorders than the Swedish-born group across all periods of follow-up, suggesting that this violation did not meaningfully affect the interpretation of our findings. Finally, our study was also limited to participants who met diagnostic criteria for a substance use disorder, and we were unable to investigate subclinical substance use in this sample.

## Comparison with previous literature

Broadly, our main finding that substance use disorders were lower amongst migrants, regardless of refugee status, is consistent with previous Swedish studies [22,27], including with respect to alcohol use disorders [23]. This appears to be a robust finding globally [22–27], sometimes even in the presence of significant trauma, suggesting that cultural norms and behaviors with respect to substance use may shape risk [46]. This argument may also explain why the effect of PTSD on substance use disorders appeared to be greater in the Swedish-born population. Nonetheless, our findings contradicted some other studies that have observed that refugees [47] and some non-refugee groups [23,48] in Sweden had higher rates of substance use disorders than the Swedish-born population, a finding we only observed with respect to cannabis use disorders in migrants from sub-Saharan Africa. An important difference between these studies and our own was that previous studies included migrant groups from different geographic regions. As we wanted to ensure valid comparisons between refugee and non-

refugee migrants, we restricted our sample to participants from regions of the world where Sweden has traditionally received substantial refugee flows. This excluded regions such as North America, Oceania, Western Europe, and Nordic countries, which were included in the aforementioned studies [23,48], and from where the incidence of substance abuse may be higher than amongst participants from regions included in the present study. We therefore suggest that it is important to consider region of origin and other sociocultural variation in substance use as a primary driver of disorder risk. Finally, in line with another study, we found that people with a diagnosis of PTSD were more at risk of developing a substance use disorders [32], although further studies should investigate the extent to which these associations may result from greater healthcare contact and diagnostic assessment.

### Interpretation

Incidence rates of substance use disorders in all groups were very high, marking this out as a major public mental health challenge irrespective of relative differences in risk between migrant and non-migrant populations. From this perspective, our results suggest that while substance abuse issues are a nontrivial concern in migrant groups, particularly amongst those with a history of PTSD, the substantial burden of these disorders affect the majority population in Sweden, where prevention efforts could be targeted. Lower rates of substance use disorders—which we observed to a similar extent in both refugee and non-refugee migrant groups from the same regions of origin—may be attributable to various factors, including the "healthy immigrant effect" or sociocultural and religious differences in attitudes and behaviors towards substance use. The healthy migrant effect suggests that people who chose to migrate are more motivated, adaptive, and often younger and healthier than the native-born population. However, such an effect may be less likely to explain lower rates of substance use disorders in refugees, although it has been posited [21] that they may benefit from additional institutional and contextual levers in their country of settlement, which provide further support against harmful behaviors [21,22].

Alternatively, lower rates of substance use disorders we observed for refugee and non-refugee migrants compared with the Swedish-born population may be attributable to cultural norms or religious views on substance use behaviors. Interestingly, our results indicated that rates of substance use disorders in migrant groups in Sweden tended to converge (i.e., increase) towards the rate in the Swedish-born population over time in a dose-response fashion, whether inspected via earlier age at migration or time lived in Sweden. These results are in line with similar observations in prevalence data on substance abuse [22] and with respect to suicide risk from a similar longitudinal cohort study in Sweden [8]. One possible interpretation of these findings is that acculturative processes lead some migrant groups to adopt Swedish health behaviors over time, here increasing their risk of being diagnosed with a substance use disorder. Alternatively, migrants who have lived longer in Sweden may be more likely to use the Swedish secondary healthcare system, due to better health literacy or fewer language barriers. Further studies are required to disentangle the multifactorial influences that may underlie these results. In doing so, we know that the immigrant/refugee paradox is not apparent for all psychiatric disorders, most notably schizophrenia and other psychotic disorders [7,15], suggesting that any stressors associated with refugee or migrant status may have differential effects on psychiatric morbidity, depending on an interplay of other contextual, individual, and neurobiological features.

Our study illustrates the substantial burden of mental health problems attributable to substance use disorders in the general population and amongst those with a history of PTSD, which is overrepresented in refugee and migrant groups. If generalizable, such high levels of psychiatric morbidity and potential convergence of rates in migrant groups over time to those

in the background population will present a fundamental public health concern for many nations across the globe.

## Supporting information

**S1 STROBE Checklist. Checklist for observational studies.**
(DOCX)

**S1 Text. Original protocol for the study.**
(DOCX)

**S1 Fig. Log-log plots of Schoenfeld residuals for hazard of any substance use disorder by migrant status in crude and adjusted models.** (A) Crude model and (B) adjusted model of log-log plots of Schoenfeld residuals by log time, following Cox proportional hazards modelling of any substance use disorder. Both panels indicate substantial departure from the proportional hazards assumptions for the non-refugee migrant group (green line), although a less severe violation of this assumption for refugee migrants (red line) relative to the Swedish-born population (blue line). See S5 Table for possible effect of bias on estimates.
(TIF)

**S1 Table. Region of origin classification and basic sample characteristics of migrant groups.**
(DOCX)

**S2 Table. Cohort characteristics by missingness status.**
(DOCX)

**S3 Table. Unadjusted and adjusted HRs for substance use disorders by migrant status in sensitivity analysis of enlarged sample ($N$ = 1,341,532), with adjustment for income.** HR, hazard ratio
(DOCX)

**S4 Table. Unadjusted and adjusted HRs by age at migration and time in Sweden for specific substance use disorders.** HR, hazard ratio
(DOCX)

**S5 Table. Comparison of AIC scores from adjusted Cox proportional hazards regression for main exposures on substance use disorder outcomes.** AIC, Akaike's Information Criterion
(DOCX)

**S6 Table. Risk of any substance use disorder in migrant subsample after mutual adjustment for all exposures and confounders.**
(DOCX)

**S7 Table. Examining evidence for violation of proportional hazards by migrant status for all substance use disorders.**
(DOCX)

**S8 Table. All-cause mortality rate by migrant status.**
(DOCX)

## Acknowledgments

A preprint of this manuscript and the Stata do-files (analytical scripts) have been made available at https://psyarxiv.com/m8b2k; doi: 10.31234/osf.io/m8b2k.

## Author Contributions

**Conceptualization:** Christina Dalman, James B. Kirkbride.

**Data curation:** Jennifer Dykxhoorn, Anna-Clara Hollander, James B. Kirkbride.

**Formal analysis:** Samantha Harris, Jennifer Dykxhoorn, James B. Kirkbride.

**Funding acquisition:** Jennifer Dykxhoorn, Anna-Clara Hollander, Christina Dalman, James B. Kirkbride.

**Investigation:** Jennifer Dykxhoorn, Christina Dalman, James B. Kirkbride.

**Methodology:** Samantha Harris, Jennifer Dykxhoorn, James B. Kirkbride.

**Project administration:** Christina Dalman, James B. Kirkbride.

**Resources:** Christina Dalman.

**Software:** James B. Kirkbride.

**Supervision:** Jennifer Dykxhoorn, James B. Kirkbride.

**Writing – original draft:** Samantha Harris, James B. Kirkbride.

**Writing – review & editing:** Samantha Harris, Jennifer Dykxhoorn, Anna-Clara Hollander, Christina Dalman, James B. Kirkbride.

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
