## [Decision Letter · Decision Letter 0]

25 Jul 2019

Dear Dr. Kirkbride,

Thank you very much for submitting your manuscript "A nationwide study of the incidence of substance use disorders in refugee and migrant groups in Sweden: a cohort study of 1.2 million people" (PMEDICINE-D-19-02064) for consideration for our upcoming special issue on substance mis/use at PLOS Medicine. 

Your paper was discussed among the editorial team, evaluated by the guest editors for the special issue, and sent to independent reviewers, including a statistical reviewer. The reviews are appended at the bottom of this email and any accompanying reviewer attachments can be seen via the link below:

[LINK]

In light of these reviews, we will not be able to accept the manuscript for publication in the journal in its current form, but we would like to invite you to submit a revised version that fully addresses the reviewers' and editors' comments. The questions raised by referee 2 about valid outcome ascertainment and possible selection seem important to us. 

You will appreciate that we cannot make a decision about publication until we have seen the revised manuscript and your response, and we expect to seek re-review by one or more of the reviewers. 

We hope to receive your revised manuscript within two weeks. Please email us (plosmedicine@plos.org) if you have any questions or concerns.

Please let me know if you have any questions. Otherwise, we will look forward to receiving your revised manuscript shortly. 

Sincerely,

Richard Turner PhD, for Philippa Berman, MBBS

rturner@plos.org

For researchers who wish to inquire about data access but are unfamiliar with Swedish, please ensure that you provide well-signposted non-author contact details. 

You may wish to restructure the title, e.g. "Substance use disorders in refugee and migrant groups in Sweden: a nationwide study of 1.2 million people". 

Please add some additional details to your abstract, e.g. quoting the number of refugees; summary demographic details for study participants; and the major countries or regions contributing refugees and/or migrants. 

Please add p values alongside CI where available.

Please avoid the phrase "refugee generating".

The final sentence of the "methods and findings" subsection of your abstract should summarize the study's main limitations. 

After your abstract, we will need to ask you to add a new and accessible "author summary" section in non-identical prose. You may find it helpful to refer to one or two recent research papers in PLOS Medicine to get a sense of the preferred style. 

Please restructure the text around lines 95-105 to create a shorter section stating the study's aims. Where you discuss expectations, please make these points earlier in the introduction, supported by references as appropriate. 

Early in your methods section, please state whether or not the study had a protocol or prespecified analysis plan, and if so attach the document as a supplementary file. Please identify any analyses that were not prespecified. 

In the discussion section of your main text, please craft a discrete paragraph or paragraphs summarizing the study's limitations. 

Please ensure that all references comply with journal style. We noted that journal names appeared to be missing for some references (e.g. references 6 and 10). Generally, publisher names can be removed (e.g. for reference 8). Referee 29 seems to lack full access details. 

We would suggest making the participant flowchart, supplementary figure 1, part of the main paper. 

Please make your STROBE checklist a separate supplementary file. Individual items should be referred to by section (e.g. "methods") and paragraph number rather than by page or line numbers, as the latter generally change in the event of publication. 

Comments from the reviewers:

*** Reviewer #1: 

The paper "A nationwide study of the incidence of substance use disorders in refugee and migrant groups in Sweden: a cohort study of 1.2 million people" examines whether risk of substance use disorders varies by region of origin and diagnosis of post-traumatic stress disorder. The authors find lower rates of substance use disorders in migrants and refugees and that a PTSD diagnosis was associated with substance use disorder. 

* The paper examines an important research question. However, I find the paper too descriptive and the results does not therefore contribute significantly to the progression of knowledge within this field of research. 

* The authors argue that one of the important contributions of the paper is that they study risks of substance use disorders by region of origin. However the region or origin groups are too crude to give any additional insight in how country specific and culture specific factors could have influenced the findings. Especially the group of migrants from "Asia, Middle East and North Africa" is extremely broad and includes many different cultures, religions and nationalities. Therefore it is difficult to make any meaningful interpretations from the results. 

* It is highly unclear why the authors decided to follow a cohort born between 1984 and 1997. These are fairly young immigrants. This should be explained much better in the paper. 

* It would have been interesting to separately to study the group of migrants arriving without parents separately.

* The findings does not really contribute to any new insights. We already know that many groups of migrants consume less alcohol and drugs. The paper would have been much more interesting if the analyses included information on duration of residence in Sweden and age at migration. Such variables could give important additional knowledge on migrants and refugees drinking patterns. 

* Some of the hypotheses are poorly explained. Why do the authors expect that refugees but not non-refugees have a higher risk of substance disorder? Why do the authors expect that those with PTSD diagnosis will have higher rates of substance disorders? What is the main explanation here? 

* It is unclear why the authors primarily suggest that the findings could be explained by the healthy migrant effect. I believe that substance use disorders are highly related to culture and religion and substance use is much lower in many countries outside Sweden. Therefore, the most likely explanation is not that migrants constitute a selection of "healthier" individuals with regard to substance use. It is rather that consumption patterns in the country of origin are lower for different reasons. 

* Undocumented migrants and refugees probably have much higher levels of substance use. This should be acknowledged or at least discussed in the paper.

* The restriction of the sample to participants from major refugee-generating regions should be explained earlier. 

*** Reviewer #2: 

There are two main problems with the methods that mean that this study's findings are difficult to interpret. 

The first is that there is no evidence cited on the validity of the particular outcomes (substance use disorders and PTSD) in the study. Are substance use disorder diagnoses valid in these registers? Does this validity extend to subcategories of substance use disorders, such as a cannabis use disorder. My impression is that these Nordic registers have good validity for severe mental illness, but this does not extend to other diagnoses, including PTSD and substance use subcategories. 

A second problem is that the authors have used predominately inpatient information to calculate prevalence rates - with outpatient diagnoses added from 2005. The problem here is that most substance use disorders are diagnosed in primary care (i.e. not outpatients) and therefore this study will have missed the majority of substance use disorders. This is borne out in the prevalence rates which are very low compared to other epidemiological surveys. In the current study, alcohol is estimated at an incidence of 208 per 100,000 person years, which seems around 10x lower than expected. The same applies for substance use disorders (which is reported at 290 per 100,000 person years). Some comparison of these incidence rates with other surveys, including the GBD would be helpful. 

These two problems may not differentially affect refugees and non-migrant refugees, but as the authors suggest in the limitations, such populations are less likely to access healthcare services. This will likely be more an issue for inpatient services, and therefore the findings of the relative differences between the groups may be biased. Furthermore, there may be more stigma in receiving a substance use disorder diagnosis, particularly alcohol (as it is against the Muslim religion and most refugees to Sweden will be from Islamic countries), which would further explain the reported differences. 

*** Reviewer #3 (statistical reviewer): 

This is a well-conducted and comprehensive study on the incidence of substance use disorders in refugee and migrant groups in Sweden. The study design, datasets, statistical methods and analyses, and presentation (tables and figures) and interpretation of results are mostly adequate and of a high standard. Missing data and proportional hazards assumption are two tricky issues but were dealt with very well by the authors in both analyses and discussion, well done! Only a few minor points needing attention:

1) Page 10, Line 217-219, apart from the crude incidence of SUD, can authors also provide age- and sex-adjusted incidence rates?

2) Can authors provide death data for the cohorts by groups? Assume very low as it's a young cohort but just to want to make sure there is no competing risk issue in applying cox models when outcomes are not all-cause mortality.

3) Page 6, Line 110, "Using linked, longitudinal Swedish register data from Psychiatry Sweden". By the way, any brief explanation or reference for "Psychiatry Sweden"?

***

[LINK]

---

## [Decision Letter · Decision Letter 1]

4 Sep 2019

Dear Dr. Kirkbride,

Thank you very much for re-submitting your manuscript "Substance use disorders in refugee and migrant groups in Sweden: a nationwide cohort study of 1.2 million people" (PMEDICINE-D-19-02064R1) for consideration at PLOS Medicine for our upcoming special issue on substance mis/use.

I have discussed the paper with editorial colleagues and our academic editor, and it was also seen again by three external reviewers. I am pleased to tell you that, provided the remaining editorial and production issues are dealt with, we expect to be able to accept the paper for publication in the journal.

[LINK]

We hope to receive your revised manuscript within one week. Please email us (plosmedicine@plos.org) if you have any questions or concerns.

Please let me know if you have any questions. Otherwise, we will look forward to receiving the revised manuscript soon. 

Kind regards,

Richard Turner PhD, for Philippa Berman, MBBS

rturner@plos.org

Requests from Editors:

Please remove Prof Dalman's contact from your data statement, as she is an author on the paper, and substitute an alternative contact who is not an author.

Please revisit the phrase "import healthier behaviours" (abstract and author summary). We suggest substituting a phrase reflecting the behaviours prevalent in migrants' countries of origin.

Around line 48, where you mention the regions of origin of migrant and refugee groups, we suggest revising the text to "Refugees' regions of origin were represented in proportions ranging from 6.0% (Eastern Europe & Russia) to 41.4% (Middle East and North Africa); Proportions of non-refugee migrants' regions of origin ranged from 11.8% (sub-Saharan Africa) to 33.7% (Middle East and North Africa).", or similar. 

At line 55, you may wish to adapt "... and for migrants from most regions-of-origin" to "and the findings did not differ substantially by migrants' region of origin.". 

At lines 57 and 87, considering the observational design of your study we ask you to substitute "was more strongly associated with", or similar, for "had more pronounced effects". Please also adapt any other instances of language ascribing causality in the ms. 

The sentence at line 59 ("Our findings highlight ... ) would seem more suitable for the "Conclusions" subsection of your abstract. 

Please adapt the final sentence of the "Methods and Findings" subsection of your abstract so that it begins "The main limitations of our study were possible differential or non-differential under-ascertainment (by migrant status) of those only seen in primary care.". Please add one further limitation to this sentence, which might focus on the unknown status of undocumented refugees and migrants, for example. 

At line 63, please start the sentence with "Our findings suggest that ..." or similar. 

At line 150, please correct "be were lower". 

Around lines 150-155, please remove the bullet points and convert this into narrative text, separating the relevant points with semicolons as needed. 

Please check journal name abbreviations in your reference list, and make any corrections needed (e.g. "CMAJ" rather than "Cmaj")

Are you able to add a URL for reference 41? Please include an accessed date if so. 

Please remove reference 44, unless this is available as a preprint; or if the paper is "in press" (if so, please send us the acceptance letter). 

Comments from Reviewers:

*** Reviewer #1: 

I am satisfied with the revisions made by the authors. I especially believe that considering age at migration and duration of residence have significantly increased the scientific impact of the paper. 

*** Reviewer #2: 

A thorough set of responses, and helpful new analyses. 

Some minor queries remain:

1. Can the authors provide some information in support of their statement that outpatient data was captured between 2001-2005? In other Swedish registry studies, it is stated that it is only captured after 2005. 

2. The authors should add a comparison of the incidence reported in this study with other studies in high income countries. Without this, it is still not clear to what extent their incidence rates are underestimates (based on lack of outpatient and primary care data). 

3. The new analyses find that those with PTSD are more likely to be diagnosed with substance use disorders. Can the authors expand on this? It is simply that those with PTSD are accessing healthcare and therefore are more likely to attract other diagnoses? 

*** Reviewer #3: 

Thanks authors for their efforts to improve the manuscript. I am satisfied with the response and the revision. No further issues needing attention.

***

[LINK]

---

## [Editor Report · Decision Letter 2]

27 Sep 2019

Dear Dr Kirkbride, 

On behalf of my colleagues and the academic editor, Dr. Margarita Alegria, I am delighted to inform you that your manuscript entitled "Substance use disorders in refugee and migrant groups in Sweden: a nationwide cohort study of 1.2 million people" (PMEDICINE-D-19-02064R2) has been accepted for publication in PLOS Medicine. 

PRODUCTION PROCESS

PRESS

PROFILE INFORMATION

Thank you again for submitting the manuscript to PLOS Medicine. We look forward to publishing it. 

Best wishes, 

Richard Turner

Senior Editor 

PLOS Medicine

plosmedicine.org